# Characterization of Neural Networks Automatically Mapped on Automotive-grade Microcontrollers

## ABSTRACT

Neural networks and Machine Learning in general, represent today one of the greatest expectations for the realization of models that can serve to determine the behavior and operation of different physical systems. Undoubtedly the calculation resources necessary for the training and the realization of the model are great especially if linked to the amount of data needed to detect the salient parameters of the model. At the same time, the models so obtained can be integrated on embedded systems, thanks to TinyML technologies, allowing to work exactly where the physical phenomena to analyze happen. In the consumer and industrial world these technologies have taken hold, and are also watched with interest by other sectors such as the automotive world. In this article we present a framework for the implementation of models based on neural networks on automotive family microprocessors, demonstrating their efficiency in two typical applications of the vehicle world: intrusion detection on the CAN bus communication network and the determination of the residual capacity of batteries for electric vehicles.

## CCS CONCEPTS

• **Computer systems organization** → **Embedded software**; • **Security and privacy** → *Network security*; • **Computing methodologies** → **Machine learning**.

## KEYWORDS

Microcontrollers, Neural Networks, LiIon Battery, Electric Vehicles, Machine Learning, embedded software, CAN networks

**ACM Reference Format:**
. 2018. Characterization of Neural Networks Automatically Mapped on Automotive-grade Microcontrollers. In *xxxxxxxxxxxxxxxxxx*. ACM, New York, NY, USA, 7 pages. https://doi.org/10.1145/1122445.1122456

## 1 INTRODUCTION

Machine Learning (ML) is revolutionizing the way we understand the world, allowing us to obtain valuable information and knowledge from large amounts of data that cannot be analyzed by the human brain. The data are those produced by us or our objects' digital behavior, that are collected by real or virtual sensors generating a big amount of information, requiring large amounts of resources in terms of processing power to be managed and analyzed. In this direction the research in recent years have investigated a wide

range of ML techniques available, for example, Neural Networks (NNs), Deep Learning (DL), clustering, Reinforcement Learning (RL), and so on. these techniques cannot be considered independently from processors, data centers and supercomputers. However, an increasingly important segment of research concerns the use of MicroController Units (MCUs). These processing entities are usually the core of everyday appliances such as vehicles, medical devices, personal gadgets, etc., where very often it is required to process information and make decisions without resorting to powerful mainframes. Undoubtedly the calculation resources necessary for the training and the realization of the model are great especially if linked to the amount of data necessary to detect the salient parameters of the model. At the same time, the models so obtained can be integrated on embedded systems, thanks to TinyML technologies, allowing to work exactly where the physical phenomena to analyze happen.

In the consumer and industrial world these technologies have taken hold, and are also watched with interest by other sectors such as the automotive world. One of the main characteristics of the automotive world is that the computational capacity must be localized on the processors present in the control units and therefore the ML models must be easily discretized and coded for these computing units. The purpose of MCU builders is to realize architectures and toolchains able to make ML algorithms more and more integrable within customer applications. It becomes important to focus that the purpose of tinyML architectures is to analyze data while they are being produced, while the entire training and education phase is delegated to more complex and high-performance systems. Therefore, in the toolchain, it's important to have the ability to incorporate models whose training takes place offline, and at the same time to have an eye on performances, having in mind that the electronic devices on board must also perform the primary functions for which they are designed, from engine control to the management of the different devices on board. In this article we present a framework for the implementation of models based on NNs on automotive family microprocessors, demonstrating their efficiency and performance on two typical applications of the vehicles onboard electronics: the estimation of data traffic on Controller Area Networks (CANs) inside the vehicle and the estimation of the remaining capacity in the case of Li-Ion batteries. In Section 3 we introduce the family of processors for the automotive industry on which experiments are carried out and the development tool needed to implement the solution on the MCU. In the following two Sections we present originally two case studies of vehicular interest that are challenging for the tiny implementation of ML. In Section 4 we present an Intrusion Detection System (IDS) on CAN bus traffic, and in Section 5 we present an algorithm for the estimation of the remaining battery capacity. In both cases, we describe the methods and present an accurate and original analysis of the complexity and

*Sigarch TinyMl'21, MArch 22–23, 2021, California, CA*
© 2018 Association for Computing Machinery.
ACM ISBN 978-1-4503-XXXX-X/18/06...$15.00
https://doi.org/10.1145/1122445.1122456

energy resources, described in Section 6. Finally, we present the conclusion in Section 7, discussing the main findings of the paper.

## 2  RELATED WORKS

Modern vehicles are becoming very complex products and dozens of Electronic Control Units (ECUs) interact to perform the most diverse functions.

**Figure 1: In-vehicle communication system example, taken from [20].**

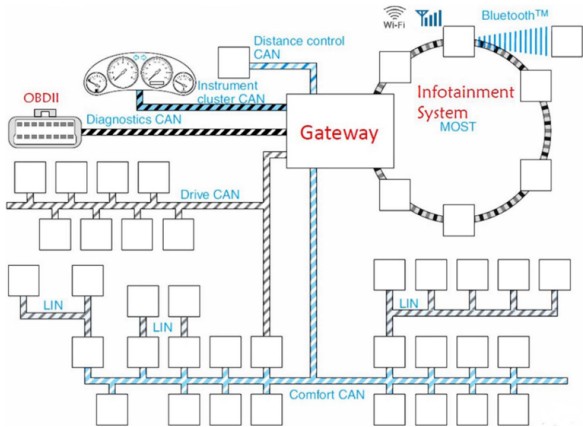

Figure 1 shows an example of an In-vehicle communication system [20] where several subnetworks divided by functionality, interconnected by means of gateway. It a microcosm of ECUs where it becomes increasingly important to use traditional control techniques combined with the most challenging techniques of ML and artificial intelligence. In automotive embedded electronics, the MCUs size and computational performance are reduced if compared with the microprocessor or processor behavior, so it is necessary to apply tiny forms of ML, already present in other consumer contexts. In the latter, the needs and requirements are very different from those required inside vehicles, which have much stricter requirements related to certifications and architectures. There are several examples in the literature of these tinyML approaches for the automotive scenario [1, 22, 26, 27]. In this paper, we want to present in a fresh way a methodology to implement ML in automotive MCUs, using the same tools used to develop traditional projects. We do on it two typical automotive applications that we describe below.

### 2.1  Intrusion Detection Systems

To have a complete picture every ECU is interconnected through a bus, the CAN bus [12], a simple but efficient solution to the problem of interconnecting ECUs that meets the requirements for real-time communication and low deployment cost. With a maximum transfer rate of 1 Mbit / sec, the CAN bus is the standard the facto for the ECU interconnections of several subnetworks that are divided by functionality and interconnected by means of gateways [20]. Car manufacturers must carefully design the interconnections between critical and no critical subnetwork trying to prevent an attacker attempt to remotely acquire the control of an ECU through a security hole, for instance in the infotainment system. It was demonstrated

[13, 24] that is possible to attack the CAN network for example disabling the braking system using the cellular connection in several vehicles. In other words, the protocol was designed in 1986 with "safety" in mind but without being secure [21]. Researchers have already found some vulnerabilities in the CAN bus:

- It is a multicast message protocol with any intrinsic mechanism of addressing and authentication. In other words, a hijacked ECU can "listen" to every message of its subnetwork and can send messages with a fake identity.
- It is a protocol with limited bandwidth for nowadays vehicles, which makes difficult the introduction of message encryption.
- Most of the nodes are automotive-grade MCUs with limited memory and computation capability and this makes difficult the implementation of complex security protocol.

The introduction of an IDS can be a countermeasure suitable for the CAN bus vulnerabilities. One of the intrusion detection methods is the anomaly-based approach. An intuitive description of this method is to consider a monitoring system, an ECU, that listens to the CAN bus traffic and learn the normal behavior. The intruder activities raise abnormal traffic, and this alerts the trained IDS. Nowadays one of the most attractive promises related to DL is the capabilities to train NNs providing a suitable amount of data with the right quality. In this paper, we propose an original version of this approach which can be embedded on MCUs.

### 2.2  Battery Residual Charge

In recent years, Lithium-Ion (Li-Ion) batteries are receiving great interest because of their several advantages in terms of high specific energy and power [23]. Rechargeable battery stacks based on Li-Ion cells are used to power many systems, including portable devices, such as smartphones, and automotive systems, such as Hybrid Electric Vehicles (HEV) and Electric Vehicles (EV) [2, 11, 28]. To increase safety, reliability, and cost-effectiveness of a battery, the performance of the Battery Management System (BMS) needs to be improved [29]. In this regard, battery capacity estimation is essential as it allows the calculation of the State of Health (SoH), e.g. a measure of battery functionality in energy storage and delivery, which is a fundamental parameter for the Battery Health Monitoring (BHM) [8]. Due to internal aging processes, capacity decays over the battery's lifetime even if it is not used, causing battery performance to decrease. Typically, a 20% reduction in rated capacity is considered the limit for safe use of the component (e.g. $C_{max} \leq 0.8 C_{rated}$), under which the battery performance may not be reliable [7, 9]. Therefore, SoH diagnosis, and accurate releasable capacity estimation, are essential for safety risks reduction, critical failure prevention, and appropriate battery replacement [10]. Data-driven methods based on ML techniques are widely used for battery capacity estimation [7]. In fact, they compute releasable capacity starting from measurable parameters, such as voltage and current, which can be easily extracted from a vehicle via CAN bus [25]. Since SoH is highly non-linear and not directly observable, DL algorithms are shown to be more flexible and efficient than traditional methods [5].

## 3 SPC5-STUDIO.AI: AUTOMATED CONVERSION OF PRE-TRAINED NEURAL NETWORKS

SPC5-Studio.AI is a plug-in component of the SPC5-STUDIO integrated development environment supporting the SPC58 "Chorus" automotive MCU family. It provides the capability to automatically generate, execute, and validate pre-trained NN models on automotive-grade MCUs. it outputs efficient "Ansi C" library that can be compiled, installed, and executed on SPC58 MCUs. DL frameworks, such as Keras, TensorFlow Lite, ONNX, Lasagne, Caffe, ConvNetJS are supported. The libraries can be integrated into the two application-specific projects defined in Sections 4 and 5, thanks to a well-defined short number of public APIs. Moreover, it provides validation and performance analysis facilities that allow to validate and characterize the converted NN and measure key metrics such as validation error, memory requirements (i.e. Flash and RAM), and execution time directly on the MCU. This plugin is integrated within SPC5-STUDIO (currently version 2.0.0).

**Figure 2: SPC5-Studio.AI block diagram**

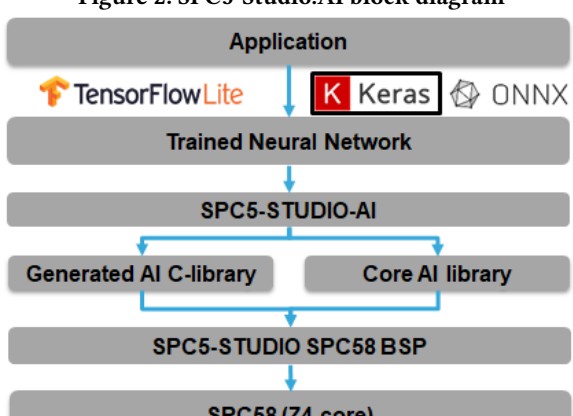

SPC5-Studio.AI was used to embed and to validate the developed NNs on three automotive-grade MCUs, suitable for applications which require low-power, connectivity, and security [31]: SPC584B, SPC58EC, and SPC58NH. The main features of the chosen MCUs are shown in Table 1. Power consumption was computed considering each MCU at its maximum frequency and with all cores enabled.

**Table 1: Main features of the automotive-grade MCUs used for the complexity analysis.**

| Device | Flash [Mb] | RAM [Mb] | Clock [MHz] | I/D Cache | FPU | Power Consumption [mA] |
|---|---|---|---|---|---|---|
| SPC584B | 2 | 192 | 120 | Yes | Yes | 102.0 |
| SPC58EC | 4 | 512 | 180 | Yes | Yes | 132.6 |
| SPC58NH | 10 | 1024 | 200 | Yes | Yes | 239.6 |

## 4 CASE STUDY 1: INTRUSION DETECTION IN AN AUTOMOTIVE NETWORK

In this Section, we present a Long Short-Term Memory (LSTM) Autoencoder to recognize CAN bus anomalies raised by abnormal traffic, using the SynCAN dataset [15]. The pre-trained Autoencoder was embedded into three different automotive MCUs via SPC5-STUDIO-AI, it was validated, and its complexity was profiled.

### 4.1 Dataset

A CAN packet consists of a timestamp, an identifier ID, and 8 bytes of payload. The packet is broadcasted on the bus and the identification field represents the type of message. The payload can carry one or more meaningful signals. Thus, the CAN bus traffic in a subnetwork can be represented by time series of signals. Figure 3 shows the structure of a CAN data frame [6].

**Figure 3: The structure of a CAN data frame, taken from [6].**

| | Arbitration | | | | | Control | Data | | CRC | | | ACK | | |
|---|---|---|---|---|---|---|---|---|---|---|---|---|---|---|
| S O F | ID | R T R | I D E | R B 0 | | DLC | DATA | | CRC | CRC DeL | A C K | ACK DeL | | E O F |

The data for the analysis were taken from SynCAN (Synthetic CAN bus data), a synthetic dataset created to benchmark, evaluate, and compare different CAN IDSs on different attack scenarios [15]. The dataset is composed of normal and abnormal traffic signals, the latter divided according to the attack type:

(1) Plateau attack. A signal overwritten to a constant value over a time period.
(2) Continuous change attack. A signal slowly drifted from its true value.
(3) Playback attack. An already recorded time series of values of the signal itself, over a time period.
(4) Suppress attack. A signal completely suppressed.
(5) Flooding attack. A signal to deny access to the other ECUs.

The anomaly detector developed was trained on the normal traffic signals, and was tested on the abnormal ones, corresponding to the attacks.

### 4.2 Long Short-Term Memory Autoencoder

The architecture used for the anomaly detector was an LSTM Autoencoder. In fact, the LSTM Autoencoder can learn the normal behavior of a simulated CAN bus traffic, as shown by [16].

The implemented Autoencoder consists of a dense layer, two LSTM layers, and a dense output layer. It is feed by 24 consecutive messages related to network traffic of 20 different signals. Thus, input data are provided in the three-dimensional format: number of samples, time steps (24), and features (20). The LSTM layers have 18 output units, and the dense output layer consists of 20 nodes. The network was made by 6272 parameters. The overall topology is shown in Table 2.

The network hyperparameters were tuned using Keras-Tuner [33]. The anomaly score was evaluated using the Mean Absolute Error (MAE) between the true network traffic and its reconstruction, made by the Autoencoder. Figure 4 shows the reconstruction error

**Table 2: The implemented Autoencoder consists of a dense layer, two LSTM layers, and a dense output layer. Input data are provided in the three-dimensional format: number of samples, time steps (24), and features (20).**

| Layer | Output shape |
|-------|--------------|
| Input | 24x20 |
| Dense | 24x20 |
| LSTM | 24x18 |
| LSTM | 24x18 |
| Dense | 24x20 |

(MAE) obtained on each attack type test set, with green representing normal CAN bus traffic and red malicious one. For all the attack types, the mean precision and recall were 0.86 and 0.81, respectively.

## 5 CASE STUDY 2: CAPACITY ESTIMATION IN LITHIUM-ION RECHARGEABLE BATTERIES

In this Section, we present a Convolutional Neural Network (CNN) LSTM architecture to predict the maximum releasable capacity of Li-Ion batteries, using the datasets made available by NASA [30]. Then, the pre-trained NN was embedded into three different automotive MCUs via SPC5-STUDIO-AI.

### 5.1 Dataset

The data were extracted from one of the Li-Ion battery datasets made available by NASA Ames Prognostics Center of Excellence (PCoE) database [30]. In all the experiments Li-Ion rechargeable batteries were run through impedance, charge, and discharge operational profiles, measuring battery impedance, temperature, voltage, current, and capacity. Due to the greater variability of discharge experimental conditions, which implies a greater complexity of the estimation, only discharge cycles have been extracted for the analysis. The battery capacity can be obtained starting from a fully charged battery and integrating the discharge current over time until it reaches a certain threshold voltage [19]. Considering that the discharge may not be complete in real-world conditions, only some samples for each discharge cycle were selected. The features used were output current, battery terminal voltage, temperature, and the time difference between samples. The targets of the prediction were the capacity values corresponding to each discharge cycle. Different batteries were used for test and training phases, thus getting closer to a real use case. During the training phase, a validation set was used to evaluate the loss function and to tune parameters (hold out method) [3].

### 5.2 Convolutional and Long Short-Term Memory Network

The architecture used for estimating maximum releasable capacity was CNN LSTM. In fact, a CNN can extract significant patterns from time series by reducing noise [4], and its temporal and spatial structure is particularly suitable for learning complex input features

[18]. Among Recurrent Neural Networks (RNNs), the LSTM architecture has been very successful with the long-term dependencies of time series [17, 32]. Moreover, while standard RNNs experience the vanishing gradient problem, LSTM networks can overcome it.

The input data are provided in the format that CNN expects, i.e. the three-dimensional one: number of samples, time steps (20), and features (4). The convolutional layer is initialized with 32 filters, of size 4x4, and it uses the ReLu activation function after output normalization. To summarize the features in the input, a max pooling layer is used for the selection of the maximum of each pair of values. Then, an LSTM layer with 32 output units [35] and with TanH activation function is used. The output is given by the dense single node output layer. The network was made by 8961 parameters. The developed CNN LSTM architecture is shown in Figure 5.

Adaptive Moment Estimation (Adam) optimization algorithm was used to train the network, and the Mean Squared Error (MSE) was the chosen loss function to minimize. Features were scaled using MinMaxScaler, which preserves the shape of the original distribution [14]. The capacity estimation error is computed using the MAE. Due to the randomness present during the training procedure (e.g. random weights initialization in NNs), at each run the results can be different. Thus, the model was trained and tested 10 times, each time using a different value for the pseudo-random number generator. The mean MAE obtained was 0.0434, below the acceptable SOH error range of ±0.05 for EVs [34]. The capacity estimation results of the CNN LSTM together with the ground truth values are shown in Figure 6. Further analysis has been made on the aforementioned dataset comparing different ML models [8], but it is out of the scope of this paper.

## 6 COMPLEXITY PROFILING

The proposed NNs were evaluated with the automotive-grade MCUs chosen: SPC584B, SPC58EC, and SPC58NH. The AI plug-in of the SPC5-STUDIO allowed the analysis of their performances in terms of Flash [Kb], Random Access Memory (RAM) [Kb], and average inference time [ms]. Tables 3 and 4 show the results obtained, for each NN.

**Table 3: Flash [Kb], RAM [Kb], and average inference time [ms] required by the LSTM Autoencoder, for each MCU chosen.**

| Device | Flash [Kb] | RAM [Kb] | Average inference time [ms] |
|--------|-----------|----------|----------------------------|
| SPC584B | 24.92 | 4.05 | 11 |
| SPC58EC | 24.92 | 4.05 | 8 |
| SPC58NH | 24.92 | 4.05 | 6 |

**Table 4: Flash [Kb], RAM [Kb], and average inference time [ms] required by the CNN LSTM, for each MCU chosen.**

| Device | Flash [Kb] | RAM [Kb] | Average inference time [ms] |
|--------|-----------|----------|----------------------------|
| SPC584B | 35.13 | 2.25 | 6.34 |
| SPC58EC | 35.13 | 2.25 | 4.38 |
| SPC58NH | 35.13 | 2.25 | 3.86 |

**Figure 4: LSTM Autoencoder reconstruction error (MAE) on each attack type test set, with green representing normal CAN bus traffic and red malicious CAN bus traffic.**

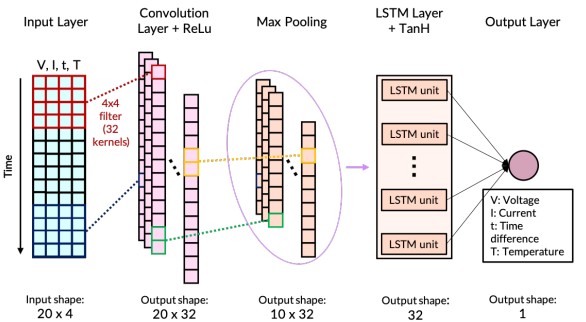

**Figure 5: The implemented CNN-LSTM Architecture consists of a 1D convolutional layer, a max pooling layer, a LSTM layer, and a dense output layer. Input data are provided in the three-dimensional format: number of samples, time steps (20), and features (4).**

**Figure 6: Capacity estimation results for CNN LSTM versus ground truth values.**

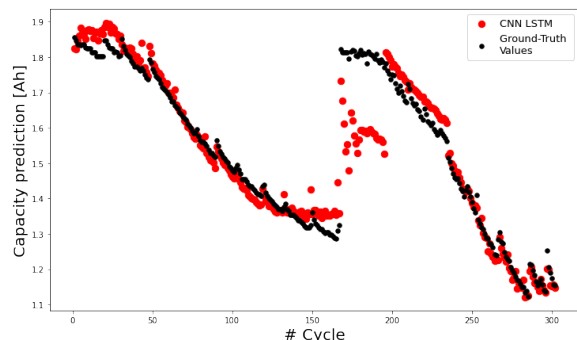

Note that only the average inference time differs between the MCUs since it decreases linearly as the clock frequency increases. Flash, RAM, and the average run time percentages are shown in Figures 7 and 8, for each layer of each model. Due to its greater

complexity, the LSTM layer is the most expensive both in terms of Flash (%), RAM (%), and average execution time (%), for both the architectures. The validation of the NNs was run on each of the chosen MCUs with 100% cross-accuracy, which uses the outputs of the original model as ground truth values for those of the C-model.

**Figure 7: Flash, RAM, and average execution time percentages obtained for each LSTM Autoencoder layer.**

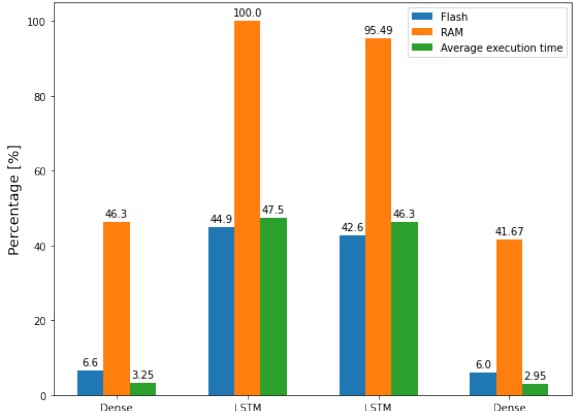

**Figure 8: Flash, RAM, and average execution time percentages obtained for each CNN LSTM layer.**

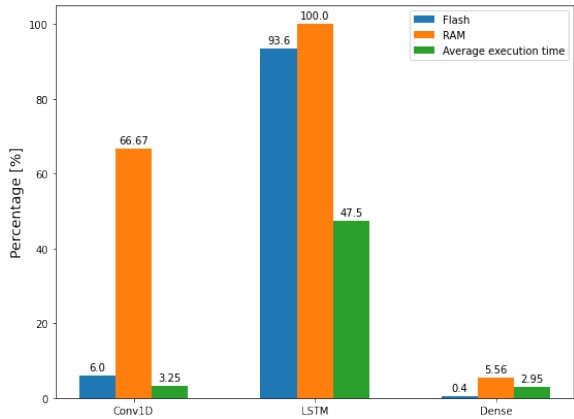

## 7 CONCLUSION

In this work, we presented the implementation through ML of two challenging problems for the automotive systems: an IDS for Can-Bus messages and an estimation method of the residual capacity in lithium batteries. The offline trained model was then quantized and made a tiny model. The two algorithms have been implemented on a family of automotive MCUs with PowerPC architecture. The accuracy of the two models is evaluated using two appropriate datasets on which the error was estimated, demonstrating the effectiveness of the two algorithms. The innovative aspect is the definition of some metrics useful to evaluate energy consumption and performance calculation of the models during their execution on the MCUs. Future work will concern the implementation of the models taking into account the safety requirements required for embedded automotive applications.

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
