# OpenReview forum: "Characterization of Neural Networks Automatically Mapped on Automotive-grade Microcontrollers"
_tinyml.org/tinyML/2021/Research_Symposium — tinyML 2021 Regular_

### Official Review · AnonReviewer1 · 2021-01-10

**Overall Merit Score:** 2

**Brief Summary:**

The paper presents the inference of two neural network models on three different automotive grade MCUs with limitted compute resources.
The first model is detecting intrusions in a CAN network, the second estimates remaining capacity of Lithium-Ion batteries.
A commercially available automated mapping tool is used (SPC5-Studio.AI).
Moreover an analysis of consumed compute resources is provided.


**Detailed Comments:**

The paper is a mix of presenting an implementation framework for neural networks on automotive MCUs, application solutions for two relevant automotive use cases and a benchmark and related metrics for an automated neural network mapping tool.
This is a very broad ambition which did not make it possible to go into sufficient depth for some relevant aspects.
In particular more detail on the mapping process, applied techniquies, related tradeoffs and their impact on the inference code would have been interesting.

**Paper Strengths:**

* The paper explains well them motivation of the work.
* It touches the very relevant topic of embedded neural computing for automotive.
* It explains the two use cases at the right level of detail.
* It provides details about the networks implemented.
* It provides figures for required memory footprint and execution times for the investigated MCUs incl. a breakdown per layer.

**Paper Weaknesses:**

* The paper does not address the specific challanges presented in automotive (e.g. what is different with the targetted MCUs?,  what is special of this work compared to mapping on an industrial or consumer MCU?, safety aspects are explicitly shifted to future work).
* The automated mapping tool is more of less used as a black box and it is discussed only very brief. It would be interesting to know what optimzations are applied for the embedded inference, which accuracy loss was seen and how accuracy was potentially regained.
* In the conclusion the definition of metrics useful to evaluate energy consumption and performance calculation is pointed out as innovative aspect of the work. However measuring memory footprint and execution latency should rather be considered state-of-the-art even when performed per network layer.


**Poster (If Paper Is Rejected):**

1: Yes, ok for poster sesion to nurture work

**Reviewer Confidence:**

4: The reviewer is confident but not absolutely certain that the evaluation is correct

---

### Official Review · AnonReviewer4 · 2021-01-27

**Overall Merit Score:** 4

**Brief Summary:**

DNN on automotive MCU, demonstrating their efficiency in two typical applications of the vehicle world:
- intrusion detection on the CAN bus communication network
- the determination of the residual capacity of batteries for electric vehicles.

**Detailed Comments:**

Easy to understand the motivation, logic and outcome.

**Paper Strengths:**

- 2 use cases are quite practically useful ones
- good explanation for In-vehicle communication system as background and the necesity of NN in CAN
- good explanation for battery charging environment
- The description of MCU specs is helpful to understand the limitation of computing
- clear benefit with clear logic and results

**Paper Weaknesses:**

- A little bit too specific to one ML compiler product, SPC5-Studio.AI. This could be explained in more general way.

**Poster (If Paper Is Rejected):**

1: Yes, ok for poster sesion to nurture work

**Reviewer Confidence:**

5: The reviewer is absolutely certain that the evaluation is correct and very familiar with the relevant literature

---

### Official Review · AnonReviewer2 · 2021-01-27

**Overall Merit Score:** 3

**Brief Summary:**

This paper presents two usecases for tiny ML for Automotive Microcontrollers :
-	Attack detection for CAN bus
-	Battery capacity estimation
In first case, an LSTM-based time series is applied. In the second, it is a combination of both LSTM and CNN.



**Detailed Comments:**

The paper shows some nice examples using Tiny ML, but it does not show any groundbreaking innovation.

Nonetheless, I propose to accept this as a presentation because it offers real-life worked examples which show that very small NNs (Tiny ML) can be used on practical use-cases (beyond the usual voice and vision cases which are often presented).


**Paper Strengths:**

The paper illustrates how small NNs can be used to solve real-life problems faced in MCUs.
- The problem of CAN attacks :
(1) Plateau attack. A signal overwritten to a constant value over
a time period.
(2) Continuous change attack. A signal slowly drifted from its
true value.
(3) Playback attack. An already recorded time series of values
of the signal itself, over a time period.
(4) Suppress attack. A signal completely suppressed.
(5) Flooding attack. A signal to deny access to the other ECUs.
This one is solved using an architecture using LSTM (6k parameters)

The problem of estimation of remaining capacity of Li-Ion battery.
This one is solved using a combination of CNN and LSTM at around 8kB

The strength of this paper is that it shows worked examples of using TinyML for real-life use-cases an achieves fairly good results with tiny models (ones what are practical to run onboard the MCU).



**Paper Weaknesses:**

The paper shows some nice examples using TinyML, but it does not show any groundbreaking innovation.

**Poster (If Paper Is Rejected):**

1: Yes, ok for poster sesion to nurture work

**Reviewer Confidence:**

5: The reviewer is absolutely certain that the evaluation is correct and very familiar with the relevant literature

---

### Official Review · AnonReviewer3 · 2021-01-29

**Overall Merit Score:** 2

**Brief Summary:**

This work focuses on deploying machine learning models on automotive-grade MCUs, with the goal of 1) detecting attacks on CAN bus messages or 2) estimating the residual capacity of lithium-ion batteries.


**Detailed Comments:**

As a research paper, this work lacks any kind of meaningful novelty. While I enjoyed learning about automotive ML use-cases, I expect research papers to meet a particular standard of innovation. I think that there are some interesting research questions that the authors could have addressed and maybe can explore in the future:

1)	How can one design specialized models for automotive MCUs?
2)	Are there special training techniques for datasets produced in an automotive environment?





**Paper Strengths:**

The main strength is that this paper addresses a real-world problem and provides a fully working solution.

**Paper Weaknesses:**


The main weaknesses are twofold:

1)	The paper lacks novelty. Both ML use-cases are straightforward applications of popular ML techniques. There is no innovation that I can see.
2)	The paper is poorly written. I had trouble reading this work and understanding what the authors mean.



**Poster (If Paper Is Rejected):**

1: Yes, ok for poster sesion to nurture work

**Reviewer Confidence:**

4: The reviewer is confident but not absolutely certain that the evaluation is correct

---

### Decision · Program_Chairs · 2021-02-05

**Decision:**

Accept (Regular)

**Comment:**

Congratulations on your paper's acceptance!

Your paper has been accepted as a full-length regular paper.

Please read the reviews carefully and make sure the concerns are addressed in your final submission.

All accepted papers will be given a slot in the TinyML Summit schedule for an oral presentation on Friday, March 26, 2021.

Camera ready instructions will follow soon. All papers will be hosted on arXiv and published papers will have the following header stamp: “Published as a conference paper at TinyML Research Symposium 2021.” The paper will also be presented on the program website.